# Singlemode-Multimode-Singlemode Fiber-Optic Interferometer Signal Demodulation Using MUSIC Algorithm and Machine Learning

Nikolai Ushakov *, Aleksandr Markvart and Leonid Liokumovich

Institute of Electronics and Telecommunications, Peter the Great St. Petersburg Polytechnic University, 195251 St. Petersburg, Russia
* Correspondence: n.ushakoff@spbstu.ru

**Abstract:** The paper is aimed at improving the efficiency of signal processing for intermode fiber-optic interferometers. To do so, we propose to use the MUSIC algorithm. It is shown that the use of traditional methods for estimating the number of signal components leads to poor operation of the MUSIC algorithm when applied to intermode interference signals. The possibility of using machine learning to estimate the number of signal components was investigated. The advantage of the proposed signal processing for demodulating the signals of an intermode interferometer over the Fourier transform has been experimentally demonstrated on the examples of simultaneous strain and curvature measurement, as well as pulse-wave sensing. The results can be also applied for processing signals of other optical-fiber sensors and multi-component signals of a different nature, for example, optical coherence tomography and radar signals.

**Keywords:** intermode interferometer; spectral interferometry; interferometry; pulse-wave sensing; machine learning; optical fiber sensor





## 1. Introduction

Optical fiber sensors (OFS) are a well-established measuring tool, demonstrating such advantages over traditional electromagnetic sensors as immunity to electromagnetic interference, absence of electric currents and voltages in sensing element, as well as small footprint, ability to multiplex many sensors in one optical fiber [1,2] and a great potential for performing distributed sensing [3]. Due to these qualities, OFS is a huge success for such applications as geophysics [4], sensing in explosive and harsh environments, biomedical diagnostics [5] and many others. Interferometric OFS can be distinguished from the other types due to their high resolution, accuracy and insensitivity to intensity changes.

Intermode optical fiber interferometers [6] are gaining considerable attention from both academia and industry due to their ability of measuring various quantities, multi-parameter sensing, small footprint and the ease of manufacture and tailoring of the sensor properties. In intermode OFS, the interference occurs between different modes, propagating in a few-mode or a multimode (MM) fiber, or between the fundamental mode and cladding modes of a singlemode (SM) fiber. The most common fiber structure in which such interference occurs is a singlemode-multimode-singlemode (SMS) structure [7], where the first SM fiber excites several modes in MM fiber, which acts as a sensing element [8,9]. After propagation in MM fiber, the excited modes accumulate phase differences due to differential mode dispersion (DMD); as a result, when the modes are captured by the second SM fiber, their interference depends on MM-fiber perturbations, affecting modes' propagation constants.

Among many approaches for interferometric OFS interrogation, spectral interferometry offers such advantages as measurement of absolute value of interferometer optical path difference (OPD), ease of interrogation of multiplexed sensors and complex multi-parameter sensing elements with several interfering waves [1,10–13]. The operating prin-

ciple of spectral interferometry is to measure the intensity of light transmitted through or reflected from an interferometer with respect to light wavelength (in other words, to measure interferometer spectrum).

SMS sensors are often interrogated by means of spectral interferometry. However, the complex shapes of interference spectra lead to complications in developing signal-processing algorithms. In the simplest cases, the demodulation of SMS signals is performed by tracking the position of a distinctive feature [14,15], such as a dip with the lowest value or a peak with the highest value. However, as with most of the semi-empirical methods, such an approach is not highly reliable and can lead to unpredictable results. In addition, since only a narrow part of the spectrum is processed, the measurement resolution of such approaches cannot reach fundamental limits. Some of the more advanced methods rely on the correlation of a measured spectrum with a reference one [16]. This way, higher measurement resolution is possible; however, unaccounted types of perturbations (for example, the bending of a strain sensor) will lead to the distortion of the measured spectrum and a consequent error in the demodulation.

Much more reliable results can be obtained in the case of the identification of interference components and the estimation of their OPDs. The first step towards this was an application of the fast-Fourier-transform (FFT)-based demodulation approach [17,18], which was initially proposed for multiplexed OFS with spectral interferometric interrogation [19–21]. However, the application of FFT for SMS signal demodulation has certain disadvantages: due to spectrum leakage [22], a cross talk between different interference components can occur, leading to demodulation errors. In addition, there is no direct way to identify and estimate the number of interference components using FFT demodulation rather than some empirical analysis of the FFT modulus.

Nevertheless, the problem of processing multi-component signals is not new and has already been successfully tackled, resulting in the development of the multiple signal classification (MUSIC) method [23,24]. The MUSIC method utilizes the analysis of a signal covariance matrix, leading to the accurate evaluation of a signal spectrum, allowing to achieve super-resolution [25,26]. A modification known as root-MUSIC [27,28] also allows one to estimate the number of signal components $U$. In fact, the estimation of the number of signal components turns out to be the crucial part of root-MUSIC algorithm: in case of under-estimation, some components of a target signal will be omitted from the processing, while over-estimation increases the unwanted influence of noise on the processing result. In order to improve the component-number estimation accuracy, various information criteria have been proposed [29–31]. However, all these methods assume strictly harmonic components and cannot be directly applied to SMS signals, in which refractive index dispersion $\Delta n(\lambda)$ causes the parasitic frequency modulation of interference components.

In the current paper, we propose a solution to the above problem based on machine learning (ML). This will be achieved by training a shallow neural network (NN) to predict a number of signal components from a distribution of a signal's covariance matrix eigenvalues. Such an ML model will, in turn, be used in the MUSIC algorithm and help correctly identify the number of interference components and find their parameters. We will also demonstrate the benefit of the proposed signal-processing approach compared with the FFT-based demodulation.

## 2. Interferometric Signal Processing

### 2.1. Intermode Interferometer Signal Model

The principle of spectral interferometry is based on unambiguous correspondence between the shape of the interferometer spectrum and interferometer OPD. Therefore, by applying proper signal processing, the absolute value of an interferometer OPD can be demodulated from the experimentally measured signal [10,32,33]. In the context of our work, considering an interference of two modes, the interference signal can be written in the form

$$S = I_1 + I_2 + 2\sqrt{I_1 I_2}\cos(2\pi \Delta n L/\lambda + \varphi), \tag{1}$$

where $I_1$ and $I_2$ are the fractions of modes' intensities that are captured by a second SM fiber; $\Delta n$ is the difference in the effective refractive indexes of interfering modes; $L$ is the length of the MM fiber section; $\lambda$ is light wavelength; and $\varphi$ is the phase shift, caused by the mode coupling effects [34]. The interference signal $S$ is measured as a function of wavelength $\lambda \in [\lambda_0 - \Delta\lambda/2; \lambda_0 + \Delta\lambda/2]$, where $\lambda_0$ is the center and $\Delta\lambda$ is the width of the spectral interval, on which the interference signal is measured.

In the first works on SMS sensors, graded-index (GI) MM fibers were used in sensing sections [7,8,35]. Since the DMD in GI MM fiber is relatively small and overlap integrals between the fundamental mode of SM fiber and most higher-order modes in GI MM fiber are almost zero, the interference signals of these sensors were basically formed by the interference of only two modes. However, later on, step-index MM fibers, specialty fiber and composite-fiber structures [13,15,36,37], mostly replaced GI MM fibers due to their higher achieved sensitivities, greater range of possibly measured quantities and possibility to use shorter sensing sections. In these cases, some number of modes $P$ ($P > 2$) is excited in the MM-fiber section and is captured by the second SM fiber; hence, the spectrum of a typical state-of-the-art SMS structure is a superposition of quasi-harmonic interference components, formed by all excited modes. As a result, interference signal can be written in a form

$$S = \sum_{p=1}^{P} I_p + \sum_{p=1}^{P} \sum_{q=1}^{P} \Bigg|_{p \neq q} \sqrt{I_p I_q} \cos\left(2\pi \Delta n_{p,q} L / \lambda + \varphi_{p,q}\right), \tag{2}$$

where it must be taken into account that $I_p$, $\Delta n_{p,q}$ and $\varphi_{p,q}$ can depend on light wavelength $\lambda$.

As can be concluded from Equation (2), a large number of interfering modes leads to a great number of interference components with different oscillation frequencies and initial phases, comprising the interference signals of SMS structures. As a result, these signals usually have very complex shapes, requiring complicated signal-processing algorithms in order to demodulate the measured quantities.

However, the use of Equation (2) may be not the most convenient way to simulate SMS interference signals. Instead, we will use the following simpler and more general multi-component model

$$S_k^v = \sum_{u=1}^{U_v} A_{u,v} \cdot \cos(2\pi(f_{u,v} + C_{u,v}/2 \cdot k) \cdot k + \theta_{u,v}) + n_k^v, \tag{3}$$

where it was taken into account that interference signal is measured in a discrete form at certain $\lambda_k$ wavelengths, so that $S_k^v$ is the $k$-th sample of $v$-th interference signal (individual signal will be referred to as $S_k$); $n_k^v$ is the $k$-th sample of $v$-th realization of additive noise; $v$ is the number of simulated signals; $v \in [1, V]$, $V$ is the number of simulated signals; $A_{u,v}$ is the amplitude of the $u$-th interference component of the $v$-th signal; $U_v$ is the number of interference components in the $v$-th signal; $f_{u,v}$ is the normalized frequency of the $m$-th interference component of the $v$-th signal; and $C_{u,v}$ is the chirp rate of the $u$-th interference component of the $v$-th signal, accounting for the DMD of the sensing fiber. $k = 1, \ldots, K$, is the number of signal samples, related to the wavelength scale as

$$\lambda_k = \lambda_0 - \Delta\lambda/2 + k \cdot \Delta\lambda/K. \tag{4}$$

Let us assume that the $u$-th interference component of the $v$-th signal is the result of the interference of the $p$-th and $q$-th modes of optical fiber. In this case, it can be shown, by expanding the argument of the cosine function in Equation (2) into Taylor series with respect to $\lambda$, that $\varphi_{u,v}$ $f_{u,v}$ and $C_{u,v}$ are related to the interference components' parameters as

$$\theta_{u,v} = 2\pi \left( L \frac{\Delta n_{p,q}^0}{\lambda_0} + L \Delta n_{p,q}^{gr} \frac{\Delta\lambda}{2} + L \Delta n_{p,q}^{GDD} \frac{\Delta\lambda^2}{4} \right) + \varphi_{p,q}, \tag{5}$$

$$f_{u,v} = L\Delta n_{p,q}^{gr} \frac{\Delta\lambda}{K} - L\Delta n_{p,q}^{GDD} \frac{\Delta\lambda^2}{K}, \tag{6}$$

$$C_{u,v} = 2L\Delta n_{p,q}^{GDD} \frac{\Delta\lambda^2}{K^2}, \tag{7}$$

where

$$\Delta n_{p,q}^{gr} = \left.\frac{d\Delta n_{p,q}}{d\lambda}\right|_{\lambda=\lambda_0} - \frac{\Delta n_{p,q}^0}{\lambda_0^2}, \tag{8}$$

$$\Delta n_{p,q}^{GDD} = \frac{1}{2\lambda_0} \left.\frac{d^2\Delta n_{p,q}}{d\lambda^2}\right|_{\lambda=\lambda_0} - \frac{1}{\lambda_0^2} \left.\frac{d\Delta n_{p,q}}{d\lambda}\right|_{\lambda=\lambda_0} + \frac{\Delta n_{p,q}^0}{\lambda_0^3} \tag{9}$$

and $\Delta n_{p,q}^0$ is the difference in refractive indexes for the $p$-th and $q$-th modes at central wavelength $\lambda = \lambda_0$. For simplicity, it will be further assumed that $\Delta n_{p,q}^{GDD}$ (and, consequently, $C_{u,v}$) values are the same for all interference components.

### 2.2. Multicomponent Signal Processing Using MUSIC Algorithm

The Root-MUSIC method allows to estimate the spectrum of complex multicomponent signals, which, as follows from Equation (6), in the considered case, is equivalent to finding the OPDs of interference components. However, in order to estimate the interference-components' properties, their number $U$ must be found first. According to theoretical analysis, the number of harmonic components $U$ can be estimated by identifying the number of a signal's covariance-matrix's eigenvalues that are different from a constant level. By this means, the corresponding covariance-matrix's eigenvectors are separated into signal and noise subspaces. Eigenvectors corresponding to eigenvalues differ from a constant value form the signal subspace and reflect the target harmonic components of the signal, while eigenvectors corresponding to eigenvalues close to a constant level form the noise subspace and reflect noisy components of the signal [25,29]. The above-mentioned constant level, close to the majority of eigenvalues, is equal to additive noise variance and, therefore, theoretically, $U$ can be estimated by directly comparing eigenvalues with the noise level (the latter can be estimated by various techniques, one of which is described in [38]). However, since, in practice, eigenvalues are estimated with singular value decomposition (SVD) [25] from only a single instance of the analyzed signal, practical distributions of eigenvalues differ from the above-described theoretical one, leading to the reduced accuracy of the theoretical criteria of signal components number.

In situations when there is no easily formalized analytical solution to the problem, but the output of the system under study $X$ can be described by a numeric model with a set of parameters (including those to be estimated) or a large amount of experimental data, obtained under well-known conditions, machine learning (ML) offers a reliable and a powerful means to achieve the result. In such cases, an inverse problem can be solved by training a computer model of some defined structure to predict the unknown parameter $\varphi$ from the given system output $X$ and known parameters $r$. Currently, ML is widely used in various tasks, including the interpretation and demodulation of OFS signals [39–42].

In each machine-learning task, the problem of generating the training dataset (consisting of the system output and known parameters) and its mapping (association of each data vector $[X, r]$ with the corresponding unknown parameters' values $\varphi$) is one of the most crucial. The ability of an ML model to generalize the results of learning is directly related to how full and representative of the practical data the training dataset is. Ideally, the distributions of the parameters $r$ and $\varphi$ in the training dataset must be the same as they are in practical situations.

In our case, the unknown parameter is the number of interference components and the processed data is an SMS spectra (the level of additive noise can be estimated prior to signal processing and be input into the ML model to improve its accuracy). Therefore, the training dataset must contain the signals of structure in Equation (3), calculated for a wide range of components numbers $U_v$, as well as components' amplitudes, frequencies and phases. Such

a dataset was prepared using the NumPy library [43] in Python. The input parameters of the function were: the number of points in simulated spectra $K$ (each $k$-th point corresponds to a different wavelength $\lambda_k$ according to Equation (4)); the maximal number of interference components $U_{\max}$ (in each $v$-th simulated interference signal, the number of components $U_v$ was chosen randomly with a uniform distribution from the interval $[1, U_{\max}]$); the range of interference components' amplitudes $[A_{\min}; A_{\max}]$; the range of the additive noise standard deviation (STD) $[\sigma_{\mathrm{n\,min}}; \sigma_{\mathrm{n\,max}}]$ (in each $v$-th simulated interference signal, the standard deviation of additive noise $\sigma_v$ was chosen randomly with a uniform distribution within the given range). The normalized frequencies of interference components $f_{u,v}$ were chosen randomly with a uniform distribution from interval $[0, 1/2]$. Initial phases $\theta_{u,v}$ were chosen randomly with a uniform distribution from interval $[0, 2\pi]$.

The generated dataset, used for NN training, contained $10^6$ interference signal realizations, each with the number of points $K = 512$ (the same as in the spectrometer used in the experimental part of the work), the maximal number of interference components in one signal $U_{\max} = 20$, the range of the interference components' amplitudes $[2 \cdot 10^{-4}, 0.2]$, the range of additive noise STD $[10^{-7}, 5 \cdot 10^{-3}]$, and chirp rate $C = 1.5 \cdot 10^{-5}$. After the signals were calculated, the eigenvalues of each signal's covariance matrix were estimated by applying SVD to a matrix $R$, containing the samples of interference signal $S_k$ in such a way that $R_{m,n} = S_{m+n-1}, m, n = 1, \ldots, K/2$. As a result, an array $\xi_m, m = 1, \ldots, K/2$ of eigenvalues was calculated for each $v$-th simulated interference signal.

### 2.3. Optimization of Neural Network Structure

Architectures and sizes of state-of-the-art neural networks differ significantly, with the simplest NNs containing several neurons and the largest deep NNs having hundreds of different layers. The number of neurons in layers also varies depending on a task. In our case of estimating the number of signal components, two problem statements can be used: classification and regression. In a classification task, the NN must be trained to distinguish between the cases of different numbers of signal components; hence, it must have $U_{\max}$ neurons in the output layer, with nonzero output of the $i$-th neuron indicating the likelihood of the analyzed signal having $i$ components (more than one neuron can have nonzero output; hence, the maximal value must be found). In the regression task, there is only one neuron in the output layer. The output numeric value of this neuron is an estimate of the number of signal components $U$ (which is not necessarily an integer and must be rounded). Obviously, the problem statement as a regression is more convenient and much more easily scalable, as full re-training is not required for an NN to learn to process signals with a greater number of components.

NN training was performed using the Scikit-Learn library [44] with MLPRegressor class. The input data consisted of the eigenvalues and an estimate of the additive noise variance, which was performed by calculating the median level of an interference signal's FFT [38]. The output data was the number of signal components. The dataset was randomly separated into training and test parts in the proportion of 80% to 20%. Since the optimal structure of the NN cannot be predicted and must be found empirically, we trained several NNs and compared their performance on the test dataset. As a compromise between the network complexity and flexibility to learn complex relations, we chose a two-layer NN. The number of neurons $N_{1,2}$ in each hidden layer was varied from 10 to 500. ReLu, tanh and sigmoid activation functions were tested. The accuracy of the model, introduced as a percentage of samples with the correctly estimated number of signal components and error RMS, found as the square root of the sum of the squares of differences between the true and estimated numbers of signal components $E_{RMS} = \sqrt{\sum(U_{est} - U_{true})^2}$ were used as performance criteria. The NNs with sigmoid activation function demonstrated the best performance, which are shown in Tables 1 and 2. The best values are marked with bold text; the five best values are marked in green text. The whole process of NN training is illustrated in a flowchart in Figure 1.

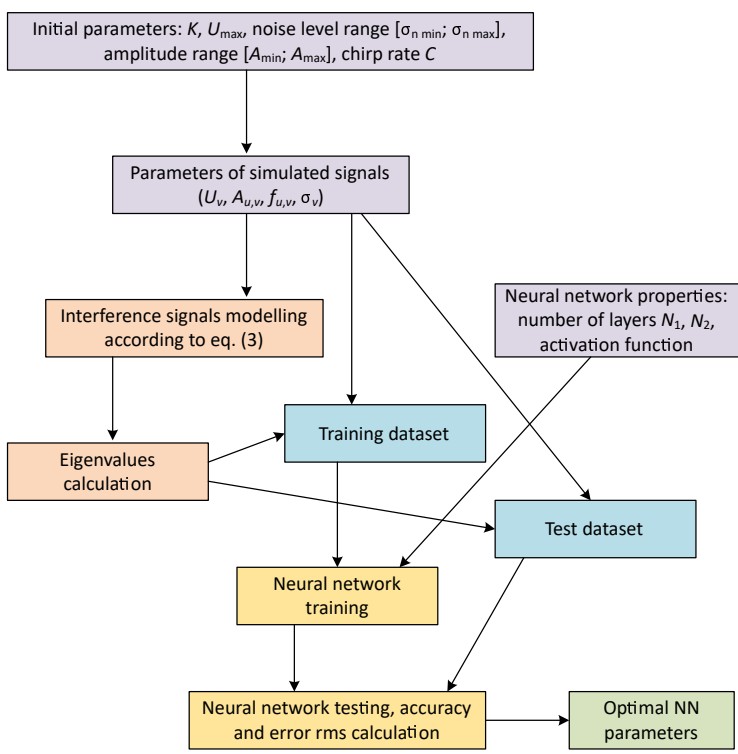

**Figure 1.** Flowchart of NN training for number of interference components estimation.

**Table 1.** NN accuracy on test dataset for different numbers of neurons in first and second layer in case of sigmoid activation function (higher values correspond to better performance).

| $N_1$ \ $N_2$ | 10 | 20 | 50 | 100 | 200 |
|---|---|---|---|---|---|
| 10 | 0.66056 | 0.66122 | 0.65592 | 0.64462 | 0.62242 |
| 20 | 0.72226 | 0.72302 | 0.71278 | 0.70728 | 0.67426 |
| 50 | 0.7622 | 0.77714 | 0.77396 | 0.7468 | 0.7586 |
| 100 | 0.77166 | 0.77106 | 0.76774 | 0.7597 | 0.77108 |
| 200 | **0.7811** | 0.77122 | 0.77998 | 0.76176 | 0.77002 |
| 500 | 0.78042 | 0.77658 | 0.7787 | 0.77078 | 0.76868 |

**Table 2.** Error RMS of NN on test dataset for different numbers of neurons in first and second layer in case of sigmoid activation function (lower values correspond to better performance).

| $N_1$ \ $N_2$ | 10 | 20 | 50 | 100 | 200 |
|---|---|---|---|---|---|
| 10 | 0.73861 | 0.74021 | 0.75137 | 0.76056 | 0.78277 |
| 20 | 0.66295 | 0.68782 | 0.67841 | 0.69287 | 0.74647 |
| 50 | 0.61353 | 0.61228 | 0.61256 | 0.64156 | 0.64039 |
| 100 | 0.61283 | 0.62669 | 0.62815 | 0.61956 | 0.62739 |
| 200 | 0.60054 | 0.61849 | 0.60925 | 0.61417 | 0.61626 |
| 500 | 0.59843 | 0.60879 | **0.59085** | 0.61704 | 0.61519 |

It can be concluded from the results presented in the tables above that the optimal NN structure incorporates 200 to 500 neurons in the first hidden layer and from 10 to 50 neurons in the second hidden layer. It can be argued which one of the calculated metrics is more important; however, since for several NN configurations both of the metrics are very close

to their best values, we chose the NN with $N_1 = 200, N_2 = 10$, which has the simplest structure among the top five in terms of both accuracy and error RMS.

The error histogram of this final NN model on test data is shown in Figure 2a. Please note the counter-clockwise rotation of the plot, which is carried out for better alignment with Figure 2b and will be explained below. The logarithmic scale of the horizontal axis is used to better visualize the probabilities of relatively large errors (the difference between the NN output and the true number of components is $|\delta U| > 1$). Figure 2b illustrates the general relation between the NN error $\delta U$ and interference signal SNR—large errors are much more likely to occur when SNR is lower than 40 dB, while for larger SNR values, most of the estimated values are correct with a relatively small fraction (11%) of small errors $|\delta U| = 1$.

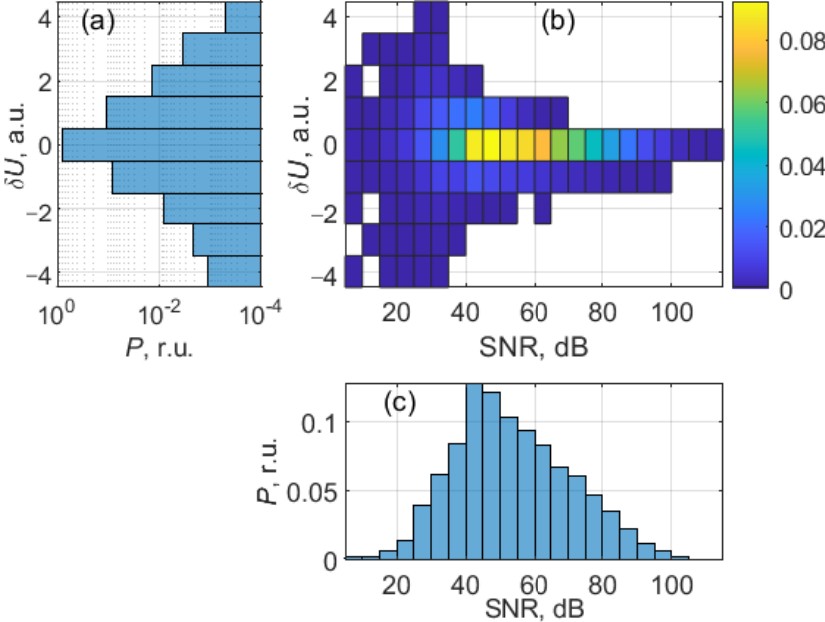

**Figure 2.** Relation of error of number of estimated components with signal-to-noise ratio. Error histogram of estimated number of components (**a**); two dimensional coincidence histogram of error and signal-to-noise ratio (**b**); signal-to-noise ratio histogram (**c**).

### 2.4. Signal Demodulation Using MUSIC Algorithm

After the number of signal components is successfully estimated by the developed ML model, optical path differences and phase increments of interference components can be found using the MUSIC algorithm. At first, interference-components' frequencies can be estimated from the noise eigenspace of the signals' covariance matrix by solving the equation [25,27].

$$a^T(z^{-1})\hat{G}\hat{G}^*a(z) = 0 \tag{10}$$

with respect to $z$, where $a(z) = [1; z^{-1}; \ldots; z^{-K/2}]$, $z = \exp(i\omega)$ and $\hat{G}$ is a matrix, whose columns contain eigenvectors corresponding to noise eigenvalues (noise eigenspace); $^T$ means matrix transpose and $^*$ is a complex conjugate. The $U$ first $z_u$ values (whose absolute values are closest to unity) are further used to find for signal demodulation. Consequently, the spatial frequencies of the interference components $\omega_u$ are found as arguments of complex values $z_u$. In turn, the initial phases of the interference components $\varphi_u$ are found equivalently to a synchronous detection by calculating correlation coefficients between the interference signal $S_k$ and complex exponents of form $\exp(i\omega_u k)$, and further calculating the arguments of these complex-valued correlation coefficients.

When the signal of a form of Equation (1) is treated as a function of wavelength $\lambda$, its oscillation frequency $\omega$ and initial phase $\varphi$ are related to the OPD according to the following equations

$$\omega = -\frac{2\pi\,\text{OPD}}{\lambda_0^2},\tag{11}$$

$$\varphi = \frac{2\pi\,\text{OPD}}{\lambda_0} - 2\pi p,\tag{12}$$

where $p$ is an integer.

Hence, in a similar manner to the work [32], interference-components' OPDs can be further found as

$$\text{OPD}_u = \frac{\lambda_0}{2}\cdot[\text{round}(\omega_u\lambda_0/\pi - \varphi_u/\pi) + \varphi_u/\pi],\tag{13}$$

where round is the rounding operation towards the nearest representative number. The whole process of interference-signal demodulation with the MUSIC algorithm is illustrated in a flowchart in Figure 3.

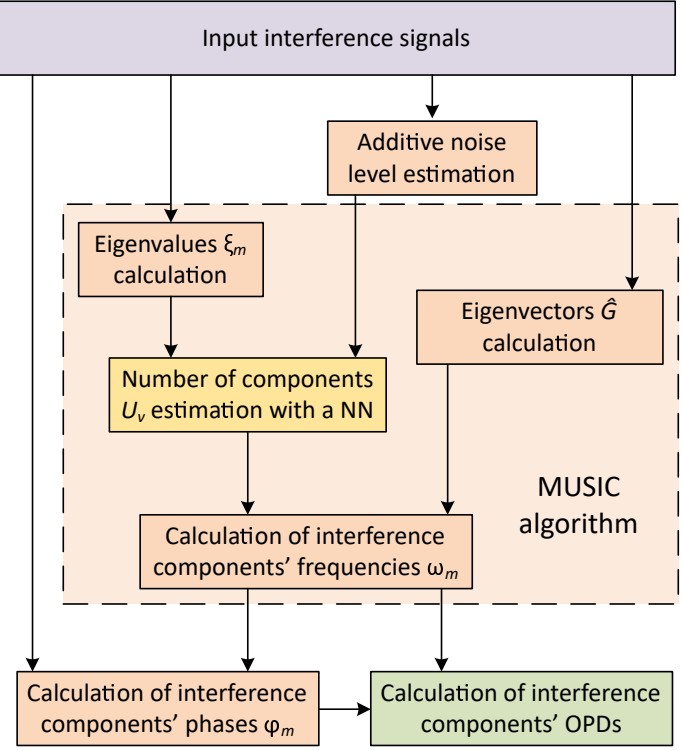

**Figure 3.** Flowchart of interference-signal demodulation.

Equation (13) allows one to find absolute values of interference-components' OPDs. It should be noted that the values obtained by Equation (13) correspond to the OPD value exactly only if there is no chromatic dispersion. This fact can be observed from Equation (6) and was extensively reported in literature [45,46].

## 3. Application of the ML-Aided MUSIC Demodulation Approach to Experimental Signal Processing

### 3.1. Experimental Measurement of Strain and Curvature Using SMS-Based Sensor

The task of multi-parameter measurements using optical fiber sensors is quite important for practical applications and has been discussed in a number of papers [13–15]. SMS-based sensors are ones of the most actively explored for this task due to several potentially independent measurands (OPDs or phase shifts of different interference components).

In this section, we will describe the application of an SMS structure for the simultaneous measurement of strain and curvature and will compare the results of FFT-based processing and MUSIC+ML processing.

The investigated SMS structure was fabricated using a standard fiber splicer Fujicura FCM 45M. The multimode section was a 32 cm long step-index Thorlabs FG050LGA fiber. At both ends, it was spliced to SMF-28 patchcords. The interrogation of the SMS sensor was performed by the NI PXIe 4844 optical sensor interrogator with a tunable laser operating in the [1.51, 1.59] μm spectral range. This device is intended for the interrogation of FBG and EFPI sensors, for which the reflective spectral characteristics are typically processed, while, in our case, transmissive spectral characteristics are to be processed. Therefore, we used a circulator to direct the light transmitted through the interferometer back to the interrogator, as shown schematically in Figure 4a.

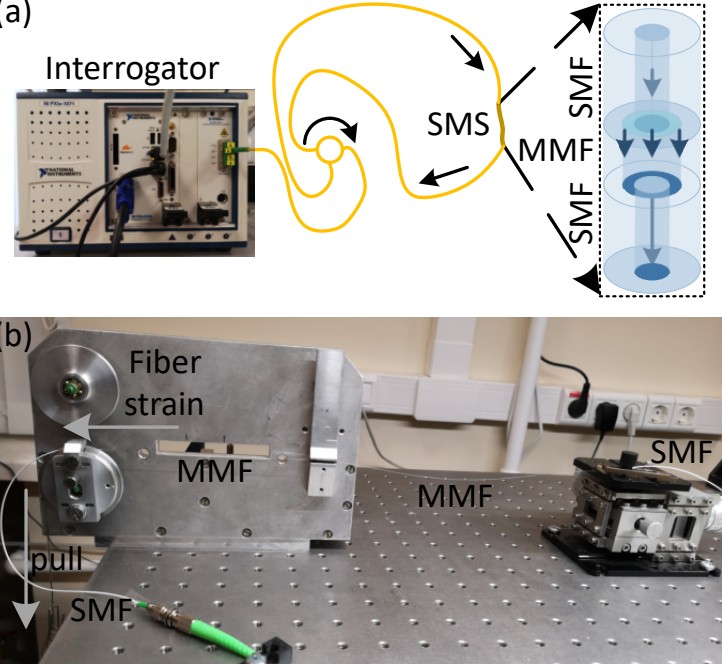

**Figure 4.** Schematic view of experimental setup (**a**) and a photograph of the experimental arrangement, used to produce a controlled simultaneous strain and bend of the sensing fiber section (**b**).

In order to subject the sensing MM section of the interferometer to simultaneous strain and curvature, we used a special experimental setup, shown in Figure 4b, in order to apply calibrated strain and bending to two distinct parts of MMF. The left part of the MMF section was subject to strain, while the right part was bent. The form of the resulting interference depends on the integral intermode phase delays, which are independent of the perturbation place. Therefore, such a setup produces conditions equivalent to both perturbations applied simultaneously to a single sensing section. Such conditions are valid while the perturbations are relatively small, yet the simplicity of the experimental setup is a great advantage in such a provisional study.

The middle of a multimode fiber was fixed by a stationary clamp; SMF on the left side was attached to a roller, to which different loads were applied, causing strain on the left part of MMF; SMF on the right side was attached to a translation stage, which was shifted to the left to increase the bend. In case of the small slack of the fiber, the bend can be assumed to be close to circular [14]. This relatively simple setup allowed to simultaneously and independently control the strain and the curvature of the two parts of the MMF section.

Thanks to the ability to separately control two types of fiber perturbation, we measured a set of experimental interference signals, corresponding to strain $S$ variation from 0 to 0.144 m$\epsilon$ (with a whole of seven uniformly spaced values) and bending curvature $C$ ranging



from 0.04 m$^{-1}$ to 0.37 m$^{-1}$ (with a whole of nine uniformly spaced values). For each pair of strain and curvature values, 100 interference signals were measured, making a total of 6300 measured signals. Examples of interference signals corresponding to $S = 0$ m$\epsilon$, $C = 0.04$ m$^{-1}$ and $S = 0.144$ m$\epsilon$, $C = 0.37$ m$^{-1}$ are shown in Figure 5.

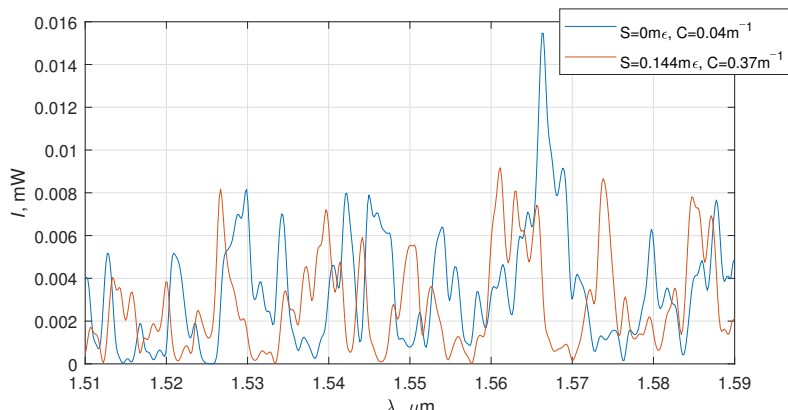

**Figure 5.** Spectral interference signals of intermode interferometer in cases of different perturbations applied to the sensing MM section.

In order to calibrate the sensor sensitivity to the perturbations and demonstrate the possibility of successful demodulation, the FFT of the measured signals were calculated; in addition, the interference-components' frequencies and phases (and, consequently, OPDs) were found using the MUSIC method. Signal subspace dimension estimation methods included the proposed ML-based approach, with the neural network having $N_1 = 200$ and $N_2 = 10$ neurons (further referred to as wide NN), as well as NN with lower accuracy, having $N_1 = 10$ and $N_2 = 10$ neurons (further referred to as narrow NN). Akaike information criterion (AIC) and minimal description length (MDL) criterion were also used for the better comparison of different methods. The Fourier transform modulus of the interference signal as well as stem plots, corresponding to estimated OPDs of interference components, are shown in Figure 6. In order to better distinguish the components found by wide and narrow neural networks, the amplitudes of the components found by narrow NN are all shown equal to 0.01. The advantage of the ML-based approach over AIC and MDL can be clearly seen in Figure 6, as it allowed the identification of most of the prominent interference components. As could be expected, the narrow NN, which demonstrated lower accuracy on simulated data, also performed worse on experimental data.

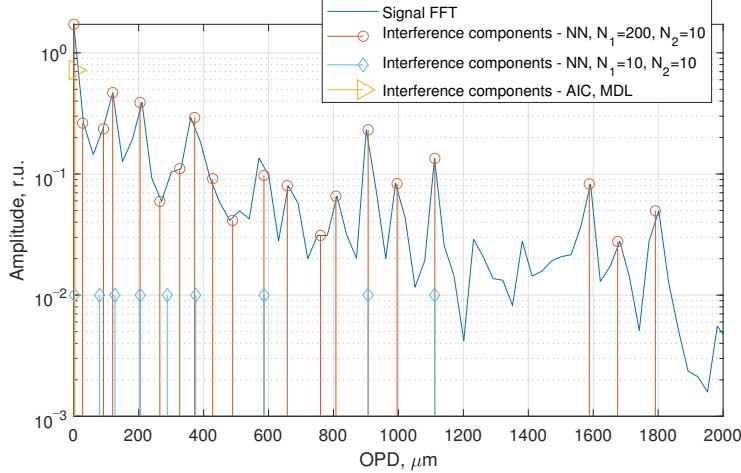

**Figure 6.** Fourier transform of the spectral interference signal shown in Figure 5 (with perturbations $S = 0$ m$\epsilon$, $C = 0.04$ m$^{-1}$) with interference components identified using the proposed ML approach and both AIC and MDL criteria.

As can be seen in Figure 6, the narrow NN turned out to be underestimating the number of interference components, leading to some errors in estimated frequencies and OPDs. Only for those components, which do not have closely situated neighbors (370 μm, 580 μm, 900 μm and 1100 μm), were frequency estimates under assumption of a different number of interference components the same, which proves the importance of accurate number of components estimation.

As can be seen from Equation (13), interference-component's frequency provides only a rough estimate of OPD, while finer precision can be gained from the phase. Therefore, phases of the most prominent interference components, listed in Table 3, measured using FFT and MUSIC methods (in case of both wide and narrow NNs), were considered for the further analysis. Experimental dependencies $\varphi(C, S)$ were approximated by plane equations using the least-squares approach. The linearity of the demodulation results was evaluated by the adjusted $R^2$ values of the fits, summarized in Table 3. The highest $R^2$ values are marked with bold font; it can be seen that the use of the MUSIC approach to demodulate the signals of the intermode interferometer allows to achieve higher response linearity. Another point, important for multiparameter sensing, is the ability to actually calculate the perturbations, which are to be measured from the phases or other parameters of the interference signal. From this point of view, it is important that different interference components have different sensitivities to different perturbations. It can be seen that the phases of interference components, demodulated by the MUSIC method, offer broader ranges of sensitivities (sensitivities with the greatest spans, corresponding to components with high linearity, are marked with bold font). In order to find the inverse relations (bend and curvature as functions of phases), constant terms $\varphi_0$ of the plane equations are also required, which were $\varphi_{0\,MW\,200\mu m} = 1.043$ and $\varphi_{0\,MW\,1000\mu m} = -0.826$ for wide NN.

For narrow NN, the best pair of components for strain and curvature demodulation is 200 μm and 1100 μm, for which the sensitivities can be found in Table 3 and constant terms are $\varphi_{0\,MN\,200\mu m} = 0.284$ and $\varphi_{0\,MN\,1100\mu m} = -1.066$. In case of FFT demodulation, the best pair of interference components are the ones with OPDs 100 μm and 1600 μm (their sensitivities are marked with italics), constant terms are $\varphi_{0\,FFT\,100\mu m} = 0.815$ and $\varphi_{0\,FFT\,1600\mu m} = 1.411$.

**Table 3.** Linearity of demodulated parameters of different interference components with respect to strain and bending of SMS structure.

| Component OPD, μm | 100 | 200 | 370 | 900 | 1000 | 1100 | 1600 | 1800 |
|---|---|---|---|---|---|---|---|---|
| $R^2_{FFT}$ | **0.995** | 0.979 | **0.98** | **0.971** | 0.968 | 0.988 | 0.993 | 0.968 |
| $\partial\varphi_{FFT}/\partial C$, rad/m$^{-1}$ | *4.1* | 5.4 | 1.5 | 0.32 | 0.26 | 6.8 | *1.2* | 8.1 |
| $\partial\varphi_{FFT}/\partial S$, rad/m$\epsilon$ | *−2.3* | 3 | 1.7 | 4.3 | 7.4 | 6.9 | *5.1* | 8.2 |
| $R^2_{MW}$ | 0.974 | **0.986** | 0.927 | 0.924 | **0.974** | **0.997** | **0.994** | **0.995** |
| $\partial\varphi_{MW}/\partial C$, rad/m$^{-1}$ | 5.5 | **6** | 9.3 | 0.9 | **−6.7** | 6.4 | 0.29 | 6.5 |
| $\partial\varphi_{MW}/\partial S$, rad/m$\epsilon$ | −4.7 | **2.7** | 13.1 | 3.8 | **9.4** | 8.1 | 8 | 10.5 |
| $R^2_{MN}$ | 0.969 | 0.982 | 0.927 | 0.924 | – | **0.997** | – | – |
| $\partial\varphi_{MN}/\partial C$, rad/m$^{-1}$ | 5.4 | 5.7 | 9.3 | 0.9 | – | 6.4 | – | – |
| $\partial\varphi_{MN}/\partial S$, rad/m$\epsilon$ | −4.5 | 2.2 | 13.1 | 3.8 | – | 8.1 | – | – |

As a result, strain and curvature can be found from the MUSIC-demodulated phases of interference components with OPDs 200 μm and 1000 μm (when wide NN was used for the number of signal components estimation) using the following matrix equation.

$$\begin{bmatrix} C \\ S \end{bmatrix} = \begin{bmatrix} 0.1886 & -0.0823 \\ -0.0562 & 0.2453 \end{bmatrix} \times \begin{bmatrix} \varphi_{MW\,200\mu m} \\ \varphi_{MW\,1000\mu m} \end{bmatrix} + \begin{bmatrix} -0.346 \\ 0.492 \end{bmatrix}. \tag{14}$$

In the case of the number of signal components estimation with narrow NN, strain and curvature can be found according to the following matrix equation

$$\begin{bmatrix} C \\ S \end{bmatrix} = \begin{bmatrix} 0.2532 & -0.0692 \\ -0.283 & 0.2387 \end{bmatrix} \times \begin{bmatrix} \varphi_{MN\,200\mu m} \\ \varphi_{MN\,1100\mu m} \end{bmatrix} + \begin{bmatrix} -0.3471 \\ 0.55 \end{bmatrix}. \tag{15}$$

In the same way, strain and curvature can be found from FFT-demodulated phases of interference components with OPDs 100 μm and 1600 μm using the following matrix equation

$$\begin{bmatrix} C \\ S \end{bmatrix} = \begin{bmatrix} 0.215 & 0.0941 \\ -0.0521 & 0.173 \end{bmatrix} \times \begin{bmatrix} \varphi_{FFT\,100\mu m} \\ \varphi_{FFT\,1600\mu m} \end{bmatrix} + \begin{bmatrix} -0.306 \\ -0.2003 \end{bmatrix}. \tag{16}$$

All fits demonstrated high linearity, with $R^2$ values of 0.987 for FFT demodulation, 0.99 for MUSIC-based demodulation with narrow NN and 0.994 for MUSIC-based demodulation with wide NN.

### 3.2. Signal Processing of a SMS-Based Pulse-Wave Sensor

Pulse-wave (PW) sensing is an important diagnostic tool, allowing to assess the state of the cardiovascular system and diagnose such diseases as hypertension, systolic heart failure, diabetes mellitus and its complications [47,48]. Optical fiber sensors offer a great solution for measuring pulse-wave signal due to their high sensitivity, multiplexing capabilities, lacking of electric currents at the sensing element and extremely small footprint, allowing their integration into smart textiles [49] and providing minimal distortion of the measured PW signal. Application of SMS sensors to PW monitoring is extremely attractive thanks to their flexibility compared to Fabry–Perot sensors and higher signal-to-noise ratio (SNR) than fiber Bragg grating sensors [50].

The investigated SMS structure was fabricated using a standard fiber splicer Fujicura FCM 45M. The MM section consisted of a 5 cm long step-index Thorlabs FG050LGA fiber. At both ends, it was spliced to SMF-28 patchcords using a Fujicura FCM 45M fiber splicer. A short length of the sensing section was chosen so that parasitic mechanical perturbations minimally affect the demodulated PW signal. However, this made the signal-processing task more challenging due to the small width of the interference signal spectrum and closely adjacent interference components. Comparing with the interferometer, studied in Section 3.1, OPDs of all interference components became 6.5 times smaller (since the same MMF and SMFs were used in the interferometer, it can be safely assumed that the structure of the interference signal will be the same); for instance, the interference component with OPD 1800 μm will have an OPD of about 280 μm.

Optical spectra were measured using an interrogation setup, consisting of a Ibsen I-MON USB512 spectrometer (spectrum measurement interval was [1.51; 1.595] μm, variable integration time from 10 μs to 100 μs, spectra acquisition rate up to 3 kHz) and Exalos EXS210066-01 SLED (output power up to 5 mW, central wavelength 1.55 μm, −6 dB spectral width 160 nm, flat-top spectrum shape, the most uniform part coincides with the spectrometer measurement range) installed on an Exalos EBD5000 driver board. A schematic illustration of the sensing setup is shown in Figure 7.

An example of a measured interference signal is shown in Figure 8a. Its Fourier transform and the interference components, estimated by the MUSIC method with different criteria of signal subspace dimension (developed ML-based approach, AIC and MDL) are shown in Figure 8b. It can be seen that despite the smaller separation of interference components' OPDs in a short SMS sensor, the proposed ML-based criterion leads to the highly accurate estimation of most of the interference components, in contrast to MDL and AIC criteria, which estimated only one signal interference component in the experimental signal.

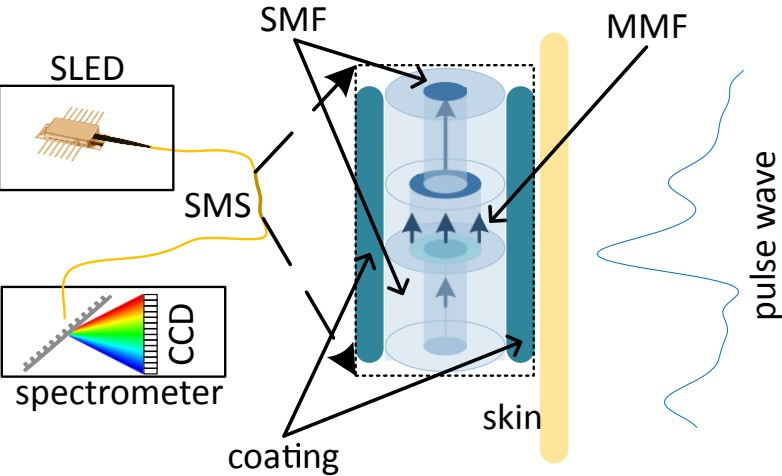

**Figure 7.** Schematic view of experimental setup.

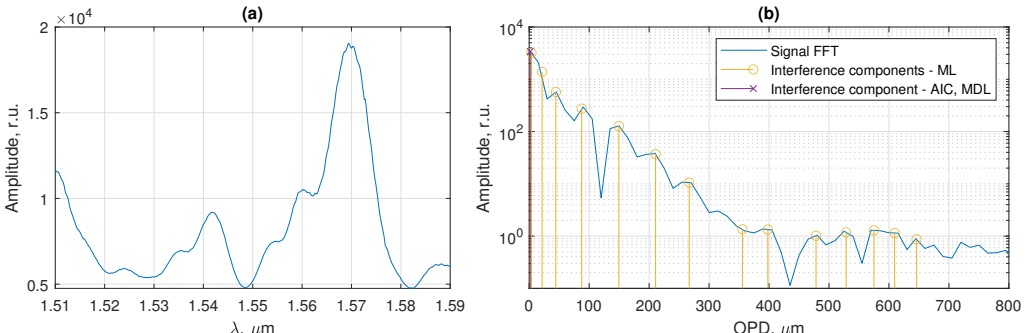

**Figure 8.** Example of a spectral-interference signal measured in the intermode interferometer with 5 cm sensing MMF section (**a**) and its Fourier transform with interference components identified using the proposed ML approach and both AIC and MDL criteria (**b**).

The SMS sensor was applied to noninvasively measure the PW signal at the radial artery. Two healthy volunteers (a man, 33 years old and a woman, 21 years old) participated in the study. Both participants were comprehensively informed about the experimental procedure and gave written consent before the experiment, which was conducted according to the Declaration of Helsinki and approved by the institutional ethics committee. PW signal was found as a phase of the interference component with the greatest amplitude. It was calculated using a conventional FFT-based demodulation algorithm, used in [50], and with a MUSIC-based algorithm as described in Section 2.4, for two cases of wide and narrow NNs being used for the number of interference components estimation. The resulting PW signals are compared in Figure 9.

It can be seen that the shape of the signal demodulated using the FFT-based algorithm is significantly distorted, as discovered in [50]; however, the signal demodulated with the MUSIC-based algorithm, when wide NN was used to estimate the number of interference components, corresponds very well to the typical PW signal shape. The PW signal feature-extraction algorithm, proposed in [51] and improved in [52], was applied to both signals to extract systolic peaks and wave feet. For the FFT-based algorithm, the obtained results were quite inaccurate, as could be anticipated, and are not shown in Figure 9. On the other hand, as can be seen in Figure 9, signal features were successfully extracted from the signal demodulated using the MUSIC-based algorithms. However, due to slight distortion of the signal, for which the number of interference components was estimated with narrow NN, wave feet were found inaccurately.

Absolute values of interference components' OPDs can be also found using the MUSIC-based algorithm and Equation (13). However, the signal shape is the same as that of the phase signal shown in Figure 9 with an ochre curve, so it is not presented here.

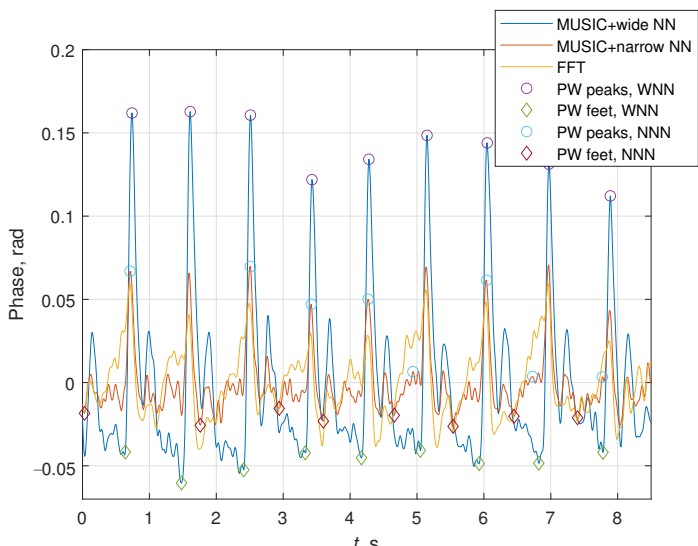

**Figure 9.** Fragments of PW signals demodulated using conventional FFT-based algorithm and the proposed MUSIC-based algorithm. For the signals demodulated using MUSIC-based algorithm (with wide and narrow NNs used for number of interference components estimation) systolic peaks and wave feet are also shown.

## 4. Conclusions

In the paper, a novel approach to processing signals of optical fiber intermode interferometer was proposed. The approach is based on the MUSIC method, widely used in radar signal processing. To adopt the MUSIC method to intermode interference signals, we proposed a signal subspace dimension estimation method, based on an artificial neural network. It was shown that the proposed approach is more efficient for multi-parameter sensing than simpler FFT-based interrogation.

By applying the proposed MUSIC and ML-based signal-processing approach to the demodulation of a pulse-wave sensor composed by a short intermode interferometer, we were able to obtain a linearized response from the sensor as compared with traditional FFT-based demodulation. Further work may include a more detailed analysis of the shape change in the pulse-wave sensor and its optimization. The proposed signal-processing approach might also be useful for improving the demodulation accuracy for other multicomponent signals, such as optical coherence tomography and multiplexed interferometers.

**Author Contributions:** Conceptualization, A.M. and N.U.; methodology, A.M. and N.U.; software, A.M. and N.U.; validation, A.M.; formal analysis, N.U.; investigation, A.M. and N.U.; resources, L.L.; data curation, A.M. and N.U.; writing—original draft preparation, A.M. and N.U.; writing—review and editing, L.L. and N.U.; visualization, A.M. and N.U.; supervision, L.L. and N.U.; project administration, L.L. and N.U.; funding acquisition, N.U. All authors have read and agreed to the published version of the manuscript.

**Funding:** Ministry of Science and Higher Education of the Russian Federation under the strategic academic leadership program "Priority 2030" (Agreement 075-15-2021-1333 dated 30 September 2021).

**Institutional Review Board Statement:** The study was conducted in accordance with the Declaration of Helsinki, and approved by the Institutional Ethics Committee of Institute of Electronics and Telecommunications (protocol code 3 dated 12 July 2022).

**Informed Consent Statement:** Informed consent was obtained from all subjects involved in the study.

**Data Availability Statement:** Obtained experimental data are available from the corresponding author upon reasonable request.

**Conflicts of Interest:** The authors declare no conflict of interest.

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
