# Peer review of "Singlemode-Multimode-Singlemode Fiber-Optic Interferometer Signal Demodulation Using MUSIC Algorithm and Machine Learning"

_photonics, doi:10.3390/photonics9110879_

Round 1

Reviewer 1 Report

This paper presents a method to use the machine learning to determine the number of signal components and combine the MUSIC algorithm to improve the accuracy of the signal demodulation of the intermode interferometer. The description is clear, and the experimental results are promising. The following questions need to be clarified before its publication.

1.      What does Mj mean in equation (3)? Please make some comments to link the variables in equation (3) to equation (2). For example, which variable in equation (2) is related to the variable fi,j?

2.      What is the definition of the error RMS in Table 2?

3.      Is the variable “OPD” in equation (5) related to the variable z in equation (4)? Please clarify this.

4.      I suggest a whole algorithm flow chart is given to help reader to better understand the propose method.

Author Response

This paper presents a method to use the machine learning to determine the number of signal components and combine the MUSIC algorithm to improve the accuracy of the signal demodulation of the intermode interferometer. The description is clear, and the experimental results are promising. The following questions need to be clarified before its publication.

  1. What does Mj mean in equation (3)? Please make some comments to link the variables in equation (3) to equation (2). For example, which variable in equation (2) is related to the variable fi,j?

Reply:

We would like to the Reviewer for the comment. We have equations (5)-(9), giving the relation between variables in eqs. (2) and (3). We have also changed some of the notations to avoid duplicating variable names.

  1. What is the definition of the error RMS in Table 2?

Reply:

We would like to the Reviewer for the comment. The definition was added in lines 177-179 of the revised manuscript.

  1. Is the variable “OPD” in equation (5) related to the variable z in equation (4)? Please clarify this.

Reply:

We would like to the Reviewer for the comment. OPD is further calculated using eq. (13) of the revised manuscript (eq. (7) in the old version), after the frequency of the interference component is found as an argument of complex value z. We have modified some comments to make it clearer.

  1. I suggest a whole algorithm flow chart is given to help reader to better understand the propose method.

Reply:

We would like to the Reviewer for the comment. The flowchart was added in figure 3.

Reviewer 2 Report

The reviewer still think that there are many problems in this manuscript.

1. The current demodulation methods can obtain the optical path difference of the interferometer, but in this manuscript, we can not see such demodulation results, which can not be compared with the existing technology, let alone the accuracy, resolution and repeatability;

2. There is also a problem compared with Fourier transform analysis, because this spectrum obviously has no periodicity and cannot be analyzed by Fourier transform.

My opinion is rejection.

Author Response

The reviewer still think that there are many problems in this manuscript.

  1. The current demodulation methods can obtain the optical path difference of the interferometer, but in this manuscript, we can not see such demodulation results, which can not be compared with the existing technology, let alone the accuracy, resolution and repeatability;

Reply:

We would like to the Reviewer for the comment. Indeed, it is a good practice to compare new methods with the already existent and thoroughly investigated ones. However, we would like to point out that FFT-based demodulation of SMS sensors was proposed a relatively long time ago and is already an accepted demodulation approach. In our work, we compare the phases demodulated by the MUSIC-based approach with the phases demodulated by the FFT-based approach and show that under controlled conditions they are in very good accordance. However, in case of SMS interferometer, comparing OPDs is much trickier, since FFT-based estimation of OPD can be either done with very low accuracy, or is suited only for Fabry-Perot sensors with either single or well separated interference components. However, we believe that adding experiments with Fabry-Perot interferometer to this manuscript will make it longer and less straightforward to follow.

  1. There is also a problem compared with Fourier transform analysis, because this spectrum obviously has no periodicity and cannot be analyzed by Fourier transform.

Authors’ reply:

We would like to the Reviewer for the comment. We would like, however, to draw Reviewer’s attention to the fact that discrete Fourier transform can be applied to signals of any shape, not necessarily only periodic. It indeed follows from the discrete Fourier transform theory that if DFT of a finite-length signal xn is treated as an integral Fourier transform of an analog signal x(t), then after inverse integral Fourier transform this x(t) signal will be a periodic repetition of the initial signal. This is a consequence of periodicity of harmonic functions, forming a basis into which the signal is decomposed. However, it does not mean that aperiodic signals can’t be analyzed using DFT, since DFT still is an unambiguous representation of any signal in a basis of harmonic components.

Examples of DFT application for aperiodic signals analysis include not only optical fiber sensors [1], but also optical coherence tomography, in which FFT is applied to signals of similar structure [2] and many others.

[1] Y. Cardona-Maya, I. Del Villar, A. B. Socorro, J. M. Corres, I. R. Matias, and J. F. Botero-Cadavid, “Wavelength and Phase Detection Based SMS Fiber Sensors Optimized with Etching and Nanodeposition,” Journal of Lightwave Technology, vol. 35, no. 17, pp. 3743–3749, Sep. 2017, doi: 10.1109/JLT.2017.2719923.

[2] B. E. Bouma et al., “Optical coherence tomography,” Nat Rev Methods Primers, vol. 2, no. 1, Art. no. 1, Oct. 2022, doi: 10.1038/s43586-022-00162-2.

My opinion is rejection.

Reply:

We thank Reviewer for reading our manuscript, but we would like to point out that the manuscript may help pave a new direction in signal processing of interference signals and demonstrates such important concepts, as advantage of applying parametric spectral estimation algorithms to interference signal processing, importance of correct estimation of number of interference components and possibility of linearizing interferometric sensor response by lowering cross-talk between interference components thanks to improved resolution of MUSIC method.

Reviewer 3 Report

In this manuscript, the authors proposed a method for processing signals of optical fiber intermode interferometer. In signal processing, MUSIC method, which is widely used in radar signal processing, is adopted. In addition, the authors used neural networks to estimate the dimension of signal subspace. The results show that the proposed method is more suitable for multi-parameter sensing than simpler FFT-based interrogation. Following is my comments/questions.

1. In order to give readers a better understanding, I suggest that the author add a flow chart of neural network training in the manuscript.

2. In lines 169 to 174, the author NN with N1 = 200, N2 = 10. How different would the results be if the author chose other values for N1 and N2? It is suggested that the authors add the results when N1 and N2 are equal to other values in the manuscript for comparison purposes.

3. The author should explain all the operators that appear in the equation, such as round () in Eq. (7).

4. I suggest authors to polish the English, a proof-reading by native speakers is required.

Author Response

In this manuscript, the authors proposed a method for processing signals of optical fiber intermode interferometer. In signal processing, MUSIC method, which is widely used in radar signal processing, is adopted. In addition, the authors used neural networks to estimate the dimension of signal subspace. The results show that the proposed method is more suitable for multi-parameter sensing than simpler FFT-based interrogation. Following is my comments/questions.

  1. In order to give readers a better understanding, I suggest that the author add a flow chart of neural network training in the manuscript.

Reply:

We would like to the Reviewer for the comment. The flowchart was added in figure 1.

  1. In lines 169 to 174, the author NN with N1 = 200, N2 = 10. How different would the results be if the author chose other values for N1 and N2? It is suggested that the authors add the results when N1 and N2 are equal to other values in the manuscript for comparison purposes.

Reply:

We would like to the Reviewer for the comment. We have added the processing results in case of N1=10, N2=10 (referred to as narrow NN in the revised manuscript) in Sections 3.1 and 3.2 of the revised manuscript.

  1. The author should explain all the operators that appear in the equation, such as round () in Eq. (7).

Reply:

We would like to the Reviewer for the comment. As suggested, all used operators were explained (lines 199 and 207).

  1. I suggest authors to polish the English, a proof-reading by native speakers is required.

Reply:

We would like to the Reviewer for the comment. We have revised the English usage thanks to a colleague who owns a Certificate of Proficiency in English. All changes are marked in the attached pdf file.

Round 2

Reviewer 2 Report

no